# (GIGA)byte

DATA RELEASE

# Chromosome-level genome assembly of the humpback puffer, *Tetraodon palembangensis*

Rui Zhang[1,†], Chang Li[1,†], Mengjun Yu[1,†], Xiaoyun Huang[1], Mengqi Zhang[1], Shanshan Liu[1], Shanshan Pan[1], Weizhen Xue[1], Congyan Wang[1], Chunyan Mao[1], He Zhang[1,2,*] and Guangyi Fan[1,*]

1   BGI-Qingdao, BGI-Shenzhen, Qingdao 266555, China
2   Department of Biology, Hong Kong Baptist University, Hong Kong, China

## ABSTRACT

The humpback puffer, *Tetraodon palembangensis*, is a poisonous freshwater pufferfish species mainly distributed in Southeast Asia (Thailand, Laos, Malaysia and Indonesia). The humpback puffer has many interesting biological features, such as inactivity, tetrodotoxin production and body expansion. Here, we report the first chromosome-level genome assembly of the humpback puffer. The genome size is 362 Mb, with a contig N50 value of ~1.78 Mb and a scaffold N50 value of ~15.8 Mb. Based on this genome assembly, ~61.5 Mb (18.11%) repeat sequences were identified, 19,925 genes were annotated, and the function of 90.01% of these genes could be predicted. Finally, a phylogenetic tree of ten teleost fish species was constructed. This analysis suggests that the humpback puffer and *T. nigroviridis* share a common ancestor 18.1 million years ago (MYA), and diverged from *T. rubripes* 45.8 MYA. The humpback puffer genome will be a valuable genomic resource to illustrate possible mechanisms of tetrodotoxin synthesis and tolerance.

**Subjects**  Animal and Plant Sciences, Animal Genetics, Functional Genomics, Marine Biology

**Submitted:**   05 November 2020

\* Corresponding authors. E-mail:
zhanghe@genomics.cn;
fanguangyi@genomics.cn

† Contributed equally.

Preprint submitted at
https://www.preprints.org/
manuscript/202008.0694/v1

## DATA DESCRIPTION

### Background and context

The humpback puffer, *Tetraodon palembangensis* (NCBI Taxonomy ID: 1820603, Fishbase ID: 25179), also known as *Pao palembangensis*, is widely distributed in Southeast Asia and prefers to live in alkalescent, warm (24–28°), and slow-flowing rivers (Figure 1a) [1]. The female and male humpback puffers have a similar body size, but the male's rear hump is much bigger than that of the female [2]. The humpback puffer is a popular ornamental fish because of its beautiful skin colouration and patterns. Unlike other species of predatory pufferfish, the humpback puffer is lazy and will not initiatively look for food [1]. Furthermore, its body contains a deadly toxin, known as tetrodotoxin (TTX), and it can swell up to three times its normal size as a defense mechanism when threatened [1]. Previous studies have shown that the toxicity of the humpback puffer varies greatly between seasons [3]. The wild population of humpback puffer has declined in recent years owing to the destruction of suitable habitat caused by climate change and overfishing [4].

With these biological characteristics and a small genome size, the humpback puffer is an ideal species for genetic study [5]. It is also a species in the Fish10K Genome Project, a subproject of the Earth BioGenome Project, which aims to sample, sequence, assemble and analyse the genomes of 10,000 fish species [6]. In this study, we provide a chromosome-scale

**Figure 1.** *Tetraodon palembangensis.* (a) Photograph of *Tetraodon palembangensis.* Photo courtesy of Preston Swipe, Aquatic Arts LLC. (b) The 17-mer depth distribution of the stLFR reads. The estimated genome size is 385 Mb; heterozygosis is 0.21%.

genome assembly of the humpback puffer. This assembly will be valuable for further study of mechanisms, such as tetrodotoxin synthesis and expansion defense. Comparative genomic analysis will help us to better understand the phylogenetic evolution and special gene families of the Tetraodontidae.

## METHODS

All methods used to isolate DNA/RNA, construct libraries, and conduct genomic sequencing are available in a protocols.io collection (Figure 2 [7]).

## Sample collection and sequencing

The sample (CNGB ID: CNS0224034) used in this study was an adult humpback puffer bought from the YueHe Flower-Bird-Fish market in Guangzhou Province, China. DNA and RNA were both isolated from blood following published protocols [8, 9]. Then, a paired-end single tube long fragment reads (stLFR) library and an RNA library were constructed according to the protocol published by Wang *et al.* [10]. A Hi-C library was constructed from blood according to the protocol published by Huang *et al.* [11]. These three libraries were then sequenced on the BGISEQ-500 platform (RRID:SCR_017979) [12]. A Nanopore library was constructed with DNA isolated from blood using the QIAamp DNA Mini Kit (Qiagen) [13] and sequenced on the GridION platform (RRID:SCR_017986) [14]. In total, we obtained 146 Gb (~312×) raw stLFR data, 21 Gb raw RNA data, 19 GB (~49×) raw Hi-C data, and 12 GB (~32×) raw Nanopore data (Table 1).

Raw stLFR reads were subjected to quality control to improve the assembly quality. Firstly, we obtained co-barcoding information from the last 42 bases of read 1 and deleted the last 42 bases. Then SOAPnuke (v1.6.5, RRID:SCR_015025) [15] was used to filter remaining reads, using the parameters: "–M 1 –d –A 0.4 –n 0.05 –l 10 –q 0.4 –Q 2 –G –5 0" [16].



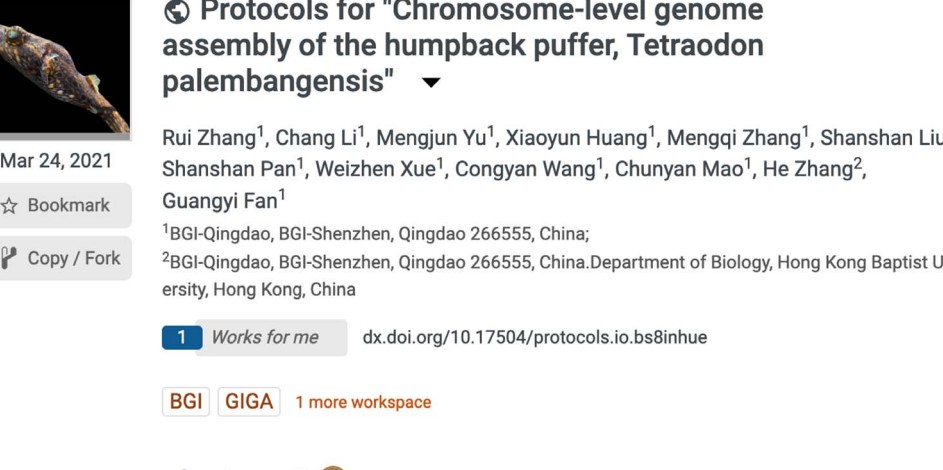

Protocols for "Chromosome-level genome assembly of the humpback puffer, Tetraodon palembangensis" ▾

Mar 24, 2021

☆ Bookmark

⚐ Copy / Fork

Rui Zhang[1], Chang Li[1], Mengjun Yu[1], Xiaoyun Huang[1], Mengqi Zhang[1], Shanshan Liu[1], Shanshan Pan[1], Weizhen Xue[1], Congyan Wang[1], Chunyan Mao[1], He Zhang[2], Guangyi Fan[1]

[1]BGI-Qingdao, BGI-Shenzhen, Qingdao 266555, China;
[2]BGI-Qingdao, BGI-Shenzhen, Qingdao 266555, China.Department of Biology, Hong Kong Baptist University, Hong Kong, China

1 *Works for me*    dx.doi.org/10.17504/protocols.io.bs8inhue

BGI   GIGA   1 more workspace

👤 zhangrui7 ⚡

**Figure 2.** Protocols.io collection for the chromosome-level genome assembly of the humpback puffer, *Tetraodon palembangensis*. https://www.protocols.io/widgets/doi?uri=dx.doi.org/10.17504/protocols.io.bs8inhue

**Table 1.** Statistics of sequencing data.

| Libraries | Read lengths (bp) | Raw data | | Valid data | |
|---|---|---|---|---|---|
| | | Total bases (Gb) | Sequencing depth (×) | Total bases (Gb) | Sequencing depth (×) |
| stFLR | PE100 | 120.4 | 311.69 | 62.6 | 162.60 |
| RNA | PE 100 | 20.6 | – | 20.0 | – |
| Hi-C | PE 100 | 19.01 | 49.43 | 5.4 | 14.04 |
| Nanopore | CN50: 32 kb | 12.3 | 31.98 | – | – |

Sequencing depth = total bases/genome size, where the genome size is the result of *k*-mer estimation in Table 2. Abbreviations: CN5, contig N50; PE, paired end.

**Table 2.** Statistics of 17-mer analysis.

| *k*-mer | *k*-mer number | *k*-mer depth (×) | Heterozygosity (%) | Genome size (bp) |
|---|---|---|---|---|
| 17 | 48,458,703,762 | 126 | 0.205 | 384,592,887 |

Finally, 62 Gb (152×) clean data were retained for further assembly. Raw RNA reads were also filtered by SOAPnuke using the parameters: "–M 1 –A 0.4 –n 0.05 –l 10 –q 0.4 –Q 2 –G –5 0", generating 20 Gb clean data. Raw Hi-C data were produced with a quality control using HiC-Pro (v. 2.8.0) [16] with default parameters. This generated 5.4 Gb validated data, which accounted for 28.81% of all data (Table 1).

## Genome assembly

Jellyfish (v2.2.6, RRID:SCR_005491) was used to count *k*-17mers of all clean stLFR reads [17]. Genomescope [18] was used to estimate the humpback puffer genome size at about 385 Mb (Table 2 and Figure 1b).

The genome size, $G$, was defined as $G = K_{num}/K_{depth}$, where the $K_{num}$ is the total number of *k*-mers, and $K_{depth}$ is the most frequently occurring frequency.

To assemble the humpback puffer genome, we firstly converted the format of clean stLFR reads, then used Supernova (v. 2.0.1, RRID:SCR_016756) to perform the draft assembly.

**Table 3.** Statistics of the draft assembly of the humpback puffer genome.

| Statistics | Scaffold | Contig |
|---|---|---|
| Total number (#) | 5291 | 6190 |
| Total length (bp) | 361,704,206 | 360,427,744 |
| Gap (N) (bp) | 1,276,462 | 0 |
| Average length (bp) | 68,362 | 58,227 |
| N50 length (bp) | 7,059,990 | 1,830,664 |
| N90 length (bp) | 453,057 | 157,209 |
| Maximum length (bp) | 19,534,197 | 9,842,180 |
| Minimum length (bp) | 682 | 48 |
| GC content (%) | 44.66 | 44.66 |

**Table 4.** Statistics of the Hi-C scaffolding of the humpback puffer genome.

| Statistics | Scaffold | Contig |
|---|---|---|
| Total number (#) | 5366 | 6435 |
| Total length (bp) | 361,698,760 | 360,427,744 |
| Gap (N) (bp) | 1,271,016 | 0 |
| Average length (bp) | 67,406 | 56,011 |
| N50 length (bp) | 15,808,960 | 1,794,775 |
| N90 length (bp) | 11,014,520 | 117,115 |
| Maximum length (bp) | 34,916,285 | 9,792,502 |
| Minimum length (bp) | 682 | 48 |
| GC content (%) | 44.66 | 44.66 |

Then, we used GapCloser (v. 1.12, RRID:SCR_015026) [19] to fill gaps with stLFR reads. To futher improve the assembly quality, TGSgapFiller [20] was then used to re-fill gaps with Nanopore reads, and Pilon (v. 1.22, RRID:SCR_014731) [21] was used to polish the assembly twice. At this stage, the genome assembly was about 362 Mb, with 7.1-Mb scaffold N50 and 1.8-Mb contig N50 values (Table 3).

With the genome and validated Hi-C data from HiC-Pro, the contact matrix was generated by Juicer (v3, RRID:SCR_017226). Finally, we perfomed chromosomal-level scaffolding using the 3D de novo assembly (3D-DNA) pipeline (v. 170123) [22]. This anchored 91.2% of all sequences to 18 chromosomes, with a length ranging from 11 Mb to 35 Mb (Table 4 and Figure 3).

The karyotype differs among genuses in Tetraodontidae [23–27]. For example, *Takifugu rubripes*, *Takifugu obscurus* and *Takifugu flavidus* have 2*n* = 44 chromosomes, while *Tetraodon nigroviridis and Tetraodon fluviatilis* have 2*n* = 42 chromosomes. In addition, *Thamnaconus septentrionalis* (Monacanthidae) has 2*n* = 40 chromosomes. Thus, we defined the chromosome number of the humpback puffer to be 18, according to the apparent and logical interactions by Hi-C reads.

## Genomic annotation

Two methods were used to annotate repetitive sequences. Firstly, we aligned the genome to the Repbase library by TRF (v.4.09) [28]. RepeatMasker (v. 3.3.0, RRID:SCR_012954) and RepeatProteinMask (v. 3.3.0) [29] were then used to predict and classify the repetitive sequences. Secondly, we constructed a repeat library using RepeatModeler (v1.0.8, RRID:SCR_015027) and classified the transposable elements (TEs) with RepeatMasker (v. 3.3.0) [29]. The results of both methods were amalgamated to give a total of 65 Mb repeat sequences and 59 Mb TEs, accounting for 18.11% (Table 5 and Figure 4a) and 16.62% of the entire genome, respectively (Table 6 and Figure 4a).

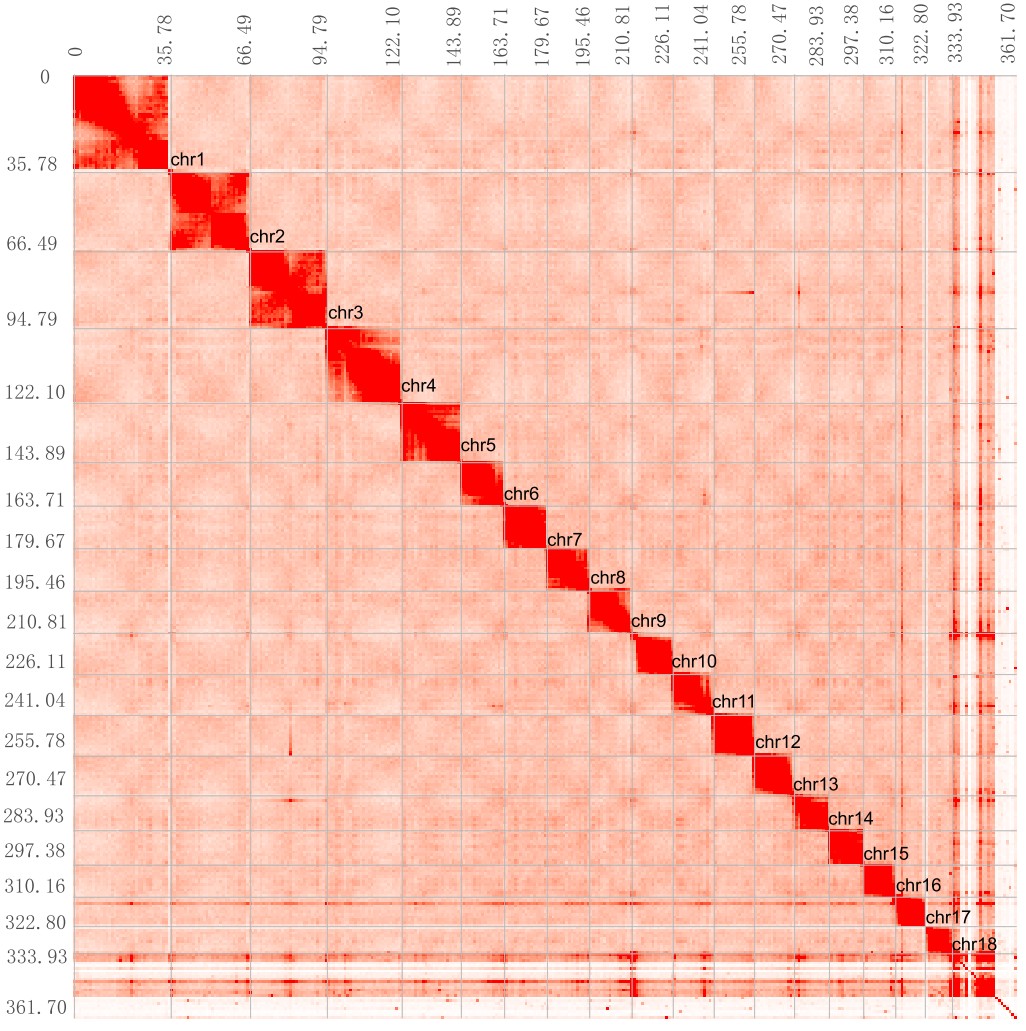

**Figure 3.** Heat map of chromosomal interaction from Hi-C reads. Grey lines show the border between chromosomes.

Using the clean, reformatted stLFR reads, the mitochondrial genome of the humpback puffer was assembled using MitoZ [30] with default parameters. Mitochondrial genes were annotated using the MitoAnnotator tool of the mitofish pipeline [31] (Figure 4b). For gene structural annotation, we performed *de novo* prediction using AUGUSTUS (v3.1, RRID:SCR_008417) [32], GlimmerHMM (v3.0.4, RRID:SCR_002654) [33], and Genscan (RRID:SCR_013362) [34]. We also used TRINITY (v2.8.5, RRID:SCR_013048) [35] to assemble a draft transcriptome with clean RNA reads, then HISAT2 (v2.1.0, RRID:SCR_015530)-StringTie (v1.3.4, RRID:SCR_016323) [36] and PASA (v2.3.3, RRID:SCR_014656)-TransDecoder (RRID:SCR_017647) [37] to predict transcripts. GeneWise (v2.4.1, RRID:SCR_015054) [38] was used for homologous annotation, with protein data obtained from the National Center for Biotechnology Information (NCBI) database for the following eight species: *Danio rerio* (NCBI, GenBank ID:50), *Cynoglossus semilaevis* (NCBI, GenBank ID:11788), *Gasterosteus aculeatus* (NCBI, GenBank ID:146), *Gadus morhua* (NCBI, GenBank ID:2661), *Larimichthys*

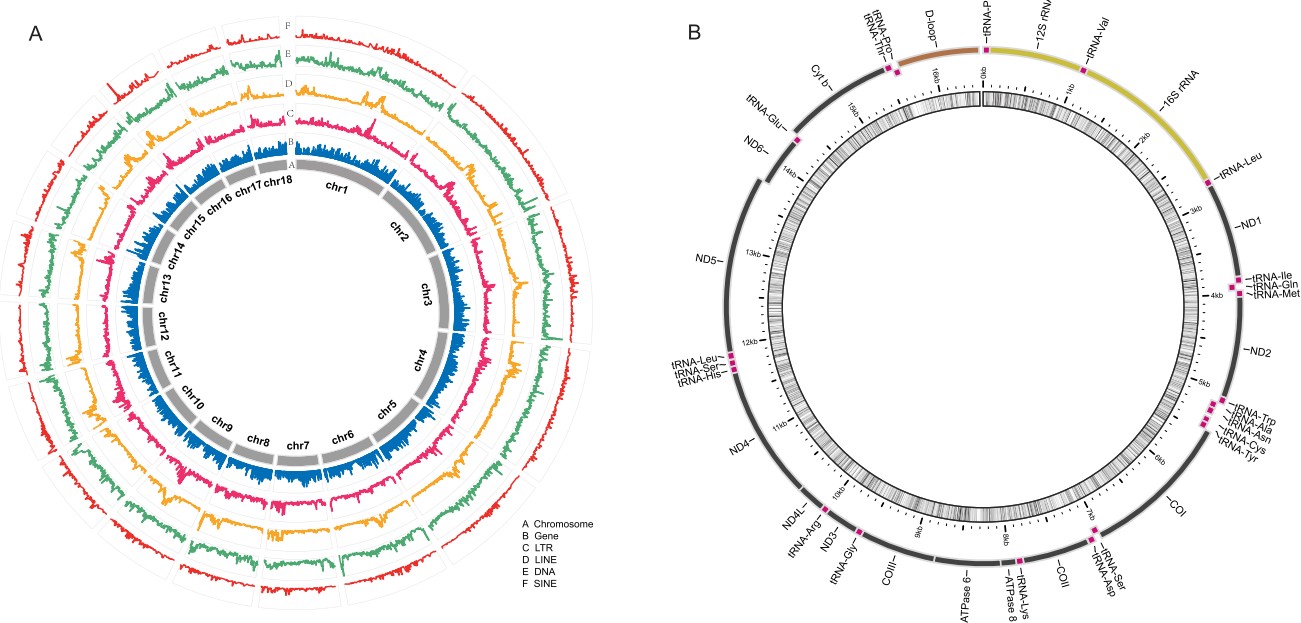

**Figure 4.** Annotation of the *Tetraodon palembangensis* genome. (a) Basic genomic elements of the genome. LTR, long terminal repeat; LINE, long interspersed nuclear elements; SINE, short interspersed elements. (b) Physical map of mitochondrial assembly.

**Table 5.** Statistics of repeat sequences.

| Type | Repeat size (bp) | % of genome |
|---|---|---|
| TRF | 9,050,571 | 2.52 |
| RepeatMasker | 34,142,529 | 9.50 |
| RepeatProteinMask | 17,674,660 | 4.92 |
| De novo | 57,492,865 | 16.00 |
| Total | 65,080,476 | 18.11 |

*crocea* (NCBI, GenBank ID:12197), *Oreochromis niloticus* (NCBI, GenBank ID:197), *Oryzias latipes* (NCBI, GenBank ID:542), and *Takifugu rubripes* (NCBI, GenBank ID:63). Finally, these three types of evidence were integrated using EVidenceModeler (v1.1.1, RRID:SCR_014659) [39], generating 19,925 nonredundant coding genes, each containing an average of 11 exons and a 1945 bp coding region (Table 7).

For gene function annotation, we aligned the 19,925 genes to the TrEMBL (UniProtKB, RRID:SCR_004426) [40], Swissprot [41], Kyoto Encyclopedia of Genes and Genomes (KEGG, RRID:SCR_012773) [42], Gene Ontology (GO, RRID:SCR_002811) [43] and InterProScan (RRID:SCR_005829) [44] databases. Overall, 90.1% of all genes were able to be functionally annotated (Table 8 and Figure 3).

## Genome evolution

To study the evolutionary status of humpback puffer among bony fish species, we clustered gene families by alignment using protein sequences of the humpback puffer and nine other teleosts (*Xiphophorus maculatus*, *Gasterosteus aculeatus*, *Sebastes schlegelii*, *Oryzias latipes*, *Gadus morhua*, *Oreochromis niloticus*, *Tetraodon nigroviridis*, *Danio rerio*, and *Takifugu rubripes*) using the TreeFam v0.50 pipeline [45]. Protein-coding genes sequences for all of



**Table 6.** Statistics of transposable elements (TEs).

| Type | RepBase TEs | | TE proteins | | *De novo* | | Combined TEs | |
|---|---|---|---|---|---|---|---|---|
| | Length (bp) | % of genome | Length (bp) | % of genome | Length (bp) | % of genome | Length (bp) | % of genome |
| DNA | 12,412,491 | 3.45 | 1,086,262 | 0.30 | 16,089,219 | 4.48 | 22,470,373 | 6.25 |
| LINE | 18,430,929 | 5.13 | 13,695,154 | 3.81 | 29,418,621 | 8.19 | 33,421,782 | 9.30 |
| SINE | 524,061 | 0.15 | 0 | 0.00 | 289,252 | 0.08 | 789,086 | 0.22 |
| LTR | 5,393,600 | 1.50 | 2,906,451 | 0.81 | 12,758,934 | 3.55 | 15,803,098 | 4.40 |
| Other | 8,290 | 0.00 | 228 | 0.00 | 0 | 0.00 | 8,518 | 0.00 |
| Unknown | 0 | 0.00 | 0 | 0.00 | 3,202,764 | 0.89 | 3,202,764 | 0.89 |
| Total | 34,142,529 | 9.50 | 17,674,660 | 4.92 | 55,052,617 | 15.32 | 59,729,335 | 16.62 |

Abbreviations: LINE, long interspersed nuclear elements; LTR, long terminal repeats; SINE, short interspersed nuclear elements; TE, transposable elements.

**Table 7.** Statistics of the predicted genes in the humpback puffer genome.

| | Gene set | Gene number | Average transcript length (bp) | Average CDS length (bp) | Average intron length (bp) | Average exon length (bp) | Average exons per gene |
|---|---|---|---|---|---|---|---|
| Homolog | *Cynoglossus semilaevis* | 19,686 | 9136.12 | 1715.14 | 856.1 | 177.4 | 9.67 |
| | *Danio rerio* | 19,348 | 15,066.80 | 1577.39 | 1718.92 | 178.28 | 8.85 |
| | *Gadus morhua* | 20,361 | 7040.85 | 1441.77 | 744.62 | 169.23 | 8.52 |
| | *Gasterosteus aculeatus* | 26,630 | 6896.85 | 1474.53 | 686.88 | 165.79 | 8.89 |
| | *Larimichthys crocea* | 21,220 | 9425.27 | 1690.06 | 902.2 | 176.53 | 9.57 |
| | *Oreochromis niloticus* | 24,562 | 9494.62 | 1789.15 | 829.18 | 173.82 | 10.29 |
| | *Oryzias latipes* | 23,332 | 8859.46 | 1467.62 | 962.5 | 169.08 | 8.68 |
| | *Takifugu rubripes* | 19,635 | 7762.47 | 1645.04 | 707.22 | 170.47 | 9.65 |
| *De novo* | Augustus | 21,662 | 7149.08 | 1725.00 | 659.42 | 186.98 | 9.23 |
| | Genscan | 25,933 | 9855.53 | 1791.43 | 990.72 | 196.01 | 9.14 |
| | GlimmerHMM | 99,722 | 1192.96 | 594.53 | 378.42 | 230.31 | 2.58 |
| Transcript | Pasa & Transdecoder | 33,965 | 4856.71 | 1186.88 | 558.33 | 156.73 | 7.57 |
| | Hisat & Stringtie | 31,664 | 5551.59 | 1303.52 | 608.39 | 163.3 | 7.98 |
| EVM | | 19,925 | 9418.80 | 1945.48 | 757.65 | 179.08 | 10.86 |

The EVM gene set contains the integrated result of *De novo* gene predictions, homolog gene predictions and transcript annotation by EVM software.

**Table 8.** Statistics of the functional annotation.

| Database | Number of genes | Gene functionally annotated (%) |
|---|---|---|
| Total | 20,057 | 100.00 |
| SwissProt | 17,333 | 86.42 |
| KEGG | 16,182 | 80.68 |
| TrEMBL | 18,037 | 89.93 |
| Interpro | 17,108 | 85.30 |
| Overall | 18,064 | 90.06 |

these species were downloaded from NCBI, except for *S. schlegelii* [46], which was obtained from the China National Genebank Nucleotide Sequence Archive (CNSA; Accession ID: CNP0000222). To improve analysis quality, we removed genes either with frameshifts, or less than 50 amino acids, as well as redundant copies, only keeping the longest transcripts for comparative genomic analysis. A total of 21,022 gene families were identified, of which 40 gene families were unique to the humpback puffer (Table 9 and Figure 5a).

Of all 21,022 gene families, we identified 4461 single-copy protein-coding genes shared by all species. We used MUSCLE v3.8.31 [47] to align these orthologs, with default parameters. Then, the alignments were concatenated into a 3,584,782 amino acid "super alignment matrix". Based on this matrix, a phylogenetic tree was constructed using RAxML v8.2.4 [48], with the best amino acid substitution model-JTT. Clade support was assessed

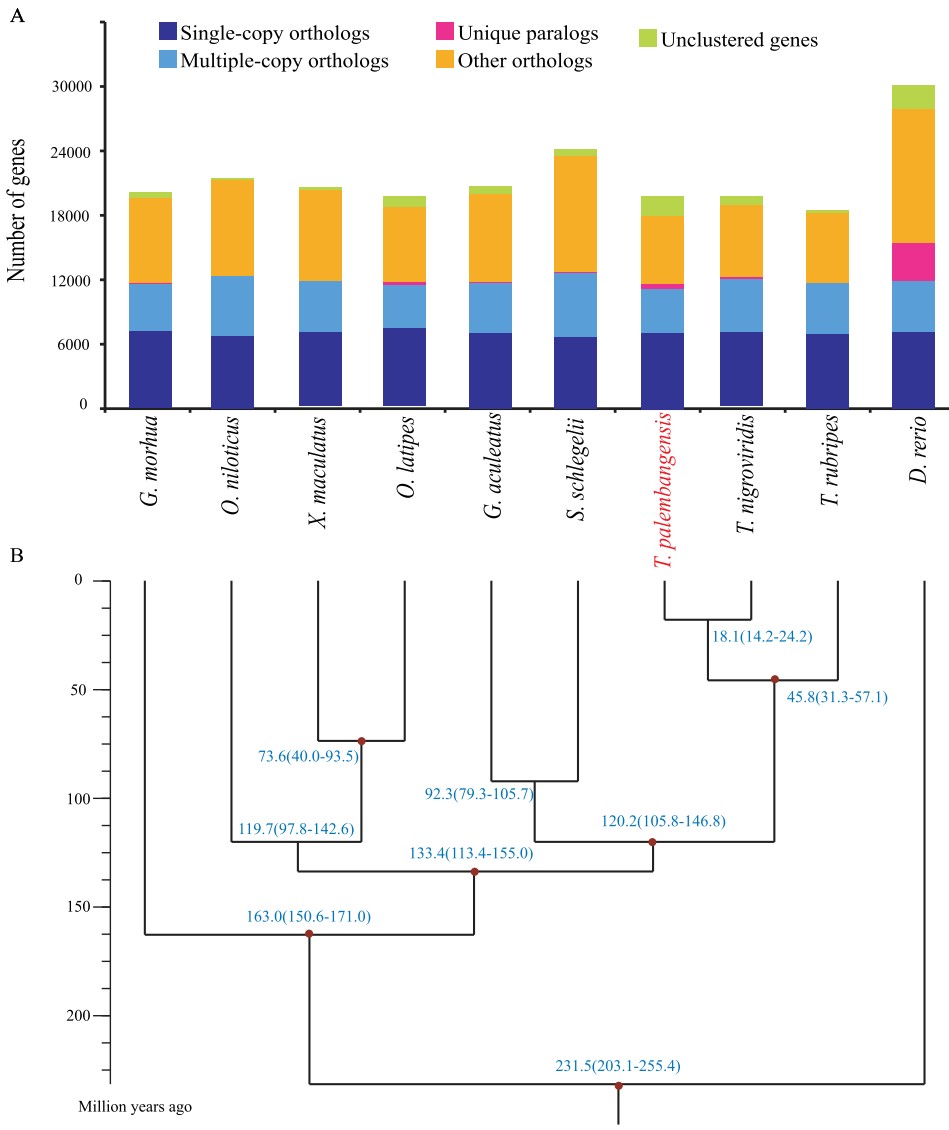

**Figure 5.** Comparative analysis of the *Tetraodon palembangensis* and nine teleosts. (a) Clustering of gene families. (b) Phylogenetic tree constructed with the single-copy gene families. The fossil correction nodes in the tree are highlighted by red dots.

using a bootstrapping algorithm with 1000 alignment replicates (Figure 5b). Next, we calculated the divergence time among these teleosts using the MCMCTree tool included in PAML (v4.7a, RRID:SCR_014932) [49], with parameters of "–rootage 500 -clock 3 -alpha 0.431879". The fossil correction time (Table 10) was obtained from Timetree [50]. The result showed that the humpback puffer and *T. nigroviridis*, two species belonging to the same genus, shared a common ancestor 18.1 millon years ago (MYA) and diverged from *T. rubripes* 18.1 MYA (Figure 5b).

**Table 9.** Statistics of gene family clustering.

| Species | Total number of genes | Number of unclustered genes | Number of gene families | Number of unique families | Average number of genes per family |
|---|---|---|---|---|---|
| *D. rerio* | 30,067 | 2171 | 18,635 | 735 | 1.5 |
| *G. aculeatus* | 20,756 | 728 | 15,995 | 11 | 1.25 |
| *G. morhua* | 19,987 | 525 | 15,650 | 11 | 1.24 |
| *S. schlegelii* | 24,094 | 558 | 16,991 | 30 | 1.39 |
| *O. latipes* | 19,535 | 984 | 14,873 | 71 | 1.25 |
| *O. niloticus* | 21,431 | 160 | 15,811 | 13 | 1.35 |
| *T. nigroviridis* | 19,544 | 805 | 14,916 | 50 | 1.26 |
| *T. palembangensis* | 19,796 | 690 | 15,830 | 40 | 1.21 |
| *T. rubripes* | 18,459 | 207 | 14,733 | 6 | 1.24 |
| *X. maculatus* | 20,356 | 271 | 16,446 | 3 | 1.22 |

**Table 10.** Fossil correction time used in divergence analysis.

| Taxon 1 | Taxon 2 | Fossil time (MYA) | Minimum (MYA) | Maximum (MYA) |
|---|---|---|---|---|
| *Takifugu rubripes* | *Tetraodon nigroviridis* | 52 | 42 | 59 |
| *Takifugu rubripes* | *Gadus morhua* | 148 | 141 | 170 |
| *Oryzias latipes* | *Xiphophorus maculatus* | 93 | 76 | 111 |
| *Oryzias latipes* | *Gasterosteus aculeatus* | 128 | 105 | 154 |
| *Oryzias latipes* | *Danio rerio* | 229.9 | 204.5 | 255.3 |

**Table 11.** Statistics of the BUSCO assessment.

| Types of BUSCOs | Genome assembly | | Gene set | |
|---|---|---|---|---|
| | Number | Percentage (%) | Number | Percentage (%) |
| Complete BUSCOs | 3486 | 95.7 | 3303 | 90.7 |
| Complete and single-copy BUSCOs | 3427 | 94.1 | 3252 | 89.3 |
| Complete and duplicated BUSCOs | 59 | 1.6 | 51 | 1.4 |
| Fragmented BUSCOs | 45 | 1.2 | 96 | 2.6 |
| Missing BUSCOs | 109 | 3.1 | 241 | 6.7 |
| Total BUSCOs groups searched | 3640 | 100 | 3640 | 100 |

## DATA VALIDATION AND QUALITY CONTROL

To demonstrate the quality of genome assembly and gene set, we performed a qulity evaluation using the actinopterygii_odb10 database from Benchmarking Universal Single-Copy Orthologs (BUSCO v.4.1.2, RRID:SCR_015008) [51]. The results showed that 95.7% and 90.7% complete BUSCOs were covered by the genome assembly and gene set, respectively (Table 11).

## REUSE POTENTIAL

We assembled the first annotated chromosome-level genome of the humpback puffer. These resources will be helpful to study the mechanism of body expansion displayed by this fish species, the synthesis mechanism and treatment of tetrodotoxin, as well as the evolution of freshwater puffer. Futhermore, the humpback puffer genome will fill a gap missing from the Fish 10K program and in the phylogenetic tree of life.

## DATA AVAILABILITY

We have deposited the project at CNGB Nucleotide Sequence Archive (CNSA) where the accession ID is CNP0001025. The genomic data can be obtained in *GigaScience* Database [52].

The sequencing data have been deposited at National Center for Biotechnology Information (NCBI) where the bioproject accession ID is PRJNA597275.

## DECLARATIONS
### LIST OF ABBREVIATIONS

bp: base pair; BUSCO: Benchmarking Universal Single-Copy Orthologs; CNSA: China National Gene Bank Nucleotide Sequence Archive; Gb: gigabase; GO: Gene Ontology; kb: kilobase; KEGG: Kyoto Enyclopedia of Genes and Genomes; Mb: megabase; ML: Maximum Likelihood; NCBI: National Center for Biotechnology Information; stLFR: single tube long fragment reads; TE: transposable element.

### ETHICAL APPROVAL

All resources used in this study were approved by the Institutional Review Board of BGI (IRB approval No. FT17007). This experiment has passed the ethics audit of Beijing Genomics Institute (BGI) Gene Bioethics and Biosecurity Review Committee.

### CONSENT FOR PUBLICATION

Not applicable.

### COMPETING INTERESTS

The authors declare that they have no competing interests.

### FUNDING

This work was supported by the special funding of "Blue granary" scientific and technological innovation of China (2018YFD0900301-05).

### AUTHORS' CONTRIBUTIONS

H.Z. and G.F. designed this project. M.Z. prepared the samples. S.L., S.P., W.X., C.W. and C.M. conducted the experiments. R.Z., C.L., M.Y. and X.H. did the analyses. R.Z., C.L. and M.Y. wrote and revised the manuscript. All authors read and approved the final version of the manuscript.

### ACKNOWLEDGEMENTS

We thank for the China National Genebank for technical support in constructing and sequencing the stLFR library.

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
