## [Reviewer Report]

Comments on revised manuscriptAfter this revision, some errors in the MS were fixed well. However, I feel it needs to be improved and revised as some errors is still existed as following:

Abstract

“Based on the genome, ~61.5Mb (18.11%) repeat sequences were identified 19,925 genes were annotated, and 90.01% of these genes could be predicted with function.”
Grammatical errors. Missing comma.

“Finally, a phylogenetic tree of ten teleost fish species was constructed .”
The result from phylogenetic analysis is still absent in the current content. This point should be improved by adding the content “Finally, a phylogenetic tree of ten teleost fish species was constructed, which suggests …”. In addition, there is a redundant space in this sentence, this kind of error should be revised throughout the whole manuscript. Please check the manuscript carefully. 

Methods 
“A Nanopore library was constructed and sequenced on the GridION platform. In total, we obtained 120 Gb (~312 X) raw stLFR data, 19 GB (~ 49X) raw Hi-C data, and 12 GB (~ 32X) raw Pacbio data” 
An obvious error present in between “Nanopore library” and “raw Pacbio data”. According to author’s response, the Pacbio data should be replaced by nanopore data.

We used Jellyfish (v2.2.6, RRID: SCR_005491) with 58 Gb clean stLFR reads to perform the k-17mer analysis [13], estimating the humpback buffer genome size about 385 Mb.
Jellyfish can count kmer amount and species easily, however, the estimation of genome size should depend on other software or script. According to Fig. S1, GenomeScope should be the source to assist the genome size estimation. However, I can not see the citation and content mentioned that.


“Finally, we perfomed the chromosomal-level scaffolding using the 3D de novo assembly (3D-DNA) pipeline (v. 170123) [17], which anchored 91.2% of total sequences to 18 chromosomes, the length ranging from 11 Mb to 35 Mb”
How define the number (18) of chromosomes should be explained as I have mentioned in previous. However, the reason is still absent. 

“Phylogenetic analyses were performed using 4,461 single-copy protein-coding genes identified by gene family analysis.”
The underline format is not required. 

Thus, although the improvement of this manuscript is obvious, yet it could not completely satisfy me, and I suggest minor revision for the current version.

---

## [Reviewer Report]

Comments on revised manuscriptThe authors amended the manuscript, yet some important concerns they did not really address. 
I really think the manuscript should be published but the lack of assembly quality control, missing information about RNA sequencing, and the way they conducted the phylogenetic analyses (super matrix rather than multi-locus) should be revised again before it can be published. 
I would like to urge the authors to have a look at these issues again, and I really do not want to be too picky but I have the feeling, that this manuscript can still be improved quite a bit.

---

## [Reviewer Report]

Upload additional filesDRR-202011-01/form/gx-DR-1604540547_SW.pdfReviewer name and names of any other individual's who aided in reviewer Sven WinterDo you understand and agree to our policy of having open and named reviews, and having your review included with the published papers. (If no, please inform the editor that you cannot review this manuscript.)YesIs the language of sufficient quality?NoPlease add additional comments on language quality to clarify if needed
In general, most of the manuscript is written in a sufficient quality, but there are certain parts that need improvement. Please see detailed comments below.Are all data available and do they match the descriptions in the paper? NoAdditional CommentsThe data under the listed BioProject PRJNA597275 is not the same as described in the manuscript. I would suggest, that the authors update the species name on NCBI and make sure that the pacbio data is uploaded (so far I only see Nanopore data). The amount of stLFR data is also more than described in the manuscript. Are the data and metadata consistent with relevant minimum information or reporting standards? See GigaDB checklists for examples <a href="http://gigadb.org/site/guide" target="_blank">http://gigadb.org/site/guide</a>YesAdditional CommentsIs the data acquisition clear, complete and methodologically sound?NoAdditional CommentsIt mostly is but it needs to be verified that pacbio and not ONT data was used. Is there sufficient detail in the methods and data-processing steps to allow reproduction?NoAdditional CommentsI would like to see more details about the library preparations. Even though the authors reference protocols, basic information, e.g., what tissue type was used for Hi-C, should be given in the manuscript. Also again, the issue with pacbio or ONT needs to be resolved, and details about either library preparation need to be specified (pacbio CLR or CCS?, ONT libprep kit and sequencer)Is there sufficient data validation and statistical analyses of data quality? YesAdditional CommentsI assume that there is, but I would like to see a bit more details of how the data was filtered, and the quality was checked. 
For example, how exactly did SOAPnuke filter the stLFR reads? What is an obvious sequencing error rate? Why was 50% of the data lost in the process? 
Same for the Hi-C data, why was there so much loss of raw data?Is the validation suitable for this type of data?YesAdditional CommentsYes I think it is, but it needs to be explained more. See detailed comments below.Is there sufficient information for others to reuse this dataset or integrate it with other data?YesAdditional CommentsAny Additional Overall Comments to the AuthorOverall, this is a good genome assembly and the manuscript is suitable for publication in Gigabyte, yet there are some methods that need to be described in more detail to be reproducible. I hope that my comments will help the authors to improve the readability and completeness of the manuscript. As page and line numbers are missing, it was easier to add my comments directly to the manuscript file. Therefore, please find my detailed comments attached.
Looking forward to seeing this manuscript published in Gigabyte soon. 
Sven Winter RecommendationMajor Revision

---

## [Reviewer Report]

Reviewer name and names of any other individual's who aided in reviewer Yunyun LvDo you understand and agree to our policy of having open and named reviews, and having your review included with the published papers. (If no, please inform the editor that you cannot review this manuscript.)YesIs the language of sufficient quality?YesPlease add additional comments on language quality to clarify if needed
Are all data available and do they match the descriptions in the paper? YesAdditional CommentsAre the data and metadata consistent with relevant minimum information or reporting standards? See GigaDB checklists for examples <a href="http://gigadb.org/site/guide" target="_blank">http://gigadb.org/site/guide</a>YesAdditional CommentsIs the data acquisition clear, complete and methodologically sound?YesAdditional CommentsIs there sufficient detail in the methods and data-processing steps to allow reproduction?YesAdditional CommentsIs there sufficient data validation and statistical analyses of data quality? NoAdditional CommentsAppropriate modificationsIs the validation suitable for this type of data?YesAdditional CommentsIs there sufficient information for others to reuse this dataset or integrate it with other data?YesAdditional CommentsAny Additional Overall Comments to the AuthorThe main contribution of this article is a new chromosome-level genomic assembly of humpback puffer with NGS, TGS and Hi-C reads. The high-quality of the assembly is reflected in the results and figures. However, I feel some results should be checked and some sentence should be rephrased until it could be officially received. I list the lines with errors or that need to be rewritten. In summary, I feel this article could be acceptable after major revisions. 

Abstract: “
Despite interesting biological features, such as its very inactive nature, tetrodotoxin production and body expansion mechanisms, molecular research on the humpback puffer is still rare because of the lack of a high-quality reference genome.” 
I feel this sentence should be rephrased. I understand a high-quality genome resource should benefit molecular research that focused on this species. However, I don’t think the reason of rare molecular studies is due to a lack of genomic assembly. In addition, “inactive nature” should be replaced by other words as my opinion. 

“scaffold N50s”?

“Based on the genome, ~61.5Mb (18.11%) repeat sequences were also identified, and totally 19,925 genes were annotated, 99.20% of which could be predicted with function using protein-coding function databases.” 
I feel this sentence should be rewritten. In addition, the inconsistence appears in the abstract and results. There are 90.1% genes could be annotated with biological function as author’s announcement in section of genome annotation. This mistake should be fixed in the revision. 

‘Finally, a phylogenetic tree was constructed with single-copy gene families from ten teleost fishes.’
This sentence just tells me you have done the phylogenetic analysis, but where the result (phylogenetic status of humpback puffer according to your analysis).

The humpback puffer genome will be a valuable genomic resource to illustrate possible mechanisms of tetrodotoxin synthesis and tolerance, providing clues for future detailed studies of biological toxins.
I understand this genome would be valuable for further studies, but I cannot see any clues for studying biological toxins in this article. 

Data Description
The author mentioned slight sex dimorphism occurred between male and female, however, the sampling sex used to genomic assembly is not described. 

‘Different from other species of predatory pufferfish, the humpback puffer is so inactive that it only moves when the food is right in front of it’
I feel confusion in this sentence. The humpback pufferfish is not predatory, or just owes lazy behavior in food seeking process?

‘Previous studies have proved that the content of the toxicity in the humpback puffer varies greatly in different seasons, so it can be edible when its skin and internal organs are removed’
I feel the former has no logical relationship with the latter.

‘In addition to these biological characteristics, the compact genome size of humpback puffer is roughly about 385 Mb, which alongside other pufferfish species which have been used to study intron evolution, makes it an ideal model species for genetic study’
This sentence is too long that bring difficulties to read.

In this study, we provided a chromosome-scale genome of an adult humpback puffer that will allow us to study features such as mechanisms of tetrodotoxin synthesis, expansion defense, body differences between males and females, and genome size. Comparative genomics analysis can help to better understand the phenotypic evolution and special gene families of the Tetraodontidae.
This sentence may mislead the readers that this article should contain those analyses related to the mentioned features, but actually not. Thus, I feel it should be rephrased. 

Methods
To be honest, I feel the section title should be “Materials, Methods and Result” as the content actually contains the three aspects.

In the assembling process, the authors assembled draft assembly with only using of stLER reads but without PB data, and the PB data was only used for closing gaps? I have not used TGSgapfiller, but I think the function of this program may be limited in gap close but not expand the assembled contigs.
Why do not use the PB reads in initial draft contig assembly?

How to define chromosomes number (18) in 3dDNA pipeline should be described. 

“1,945 bps coding region” should be “1,945 bp ”

As mentioned above, the inconsistence appears in this section and Abstract

‘We firstly used MUSCLE…..’ Type error. There is a redundant space between ‘to’ and ‘align’ in this sentence. 

The author should describe how many fossils used to correct the phylogenetic topology and which nodes be corrected. In addition, I am curious that why parameter of alpha is 0.431879. 
RecommendationMajor Revision